

# Microbiota composition and correlations with environmental factors in grass carp (*Ctenopharyngodon idella*) culture ponds in South China

Yingli Lian[1,2,3], Xiafei Zheng[4], Shouqi Xie[2], Dan A[5], Jian Wang[1,3], Jiayi Tang[1,3], Xuan Zhu[3] and Baojun Shi[1,3]

[1] Key Laboratory of Microecological Resources and Utilization in Breeding Industry, Ministry of Agriculture and Rural Affairs, Guangzhou, Guangdong, China
[2] Institute of hydrobiology, Chinese Academy of Sciences, Wuhan, Hubei, China
[3] Guangdong Haid Group Co., Ltd, Guangzhou, Guangdong, China
[4] Ninghai Institute of Mariculture Breeding and Seed Industry, Zhejiang Wanli University, Ningbo, Zhejiang, China
[5] College of Resources and Environment, Zhongkai University of Agriculture and Engineering, Guangzhou, Guangdong, China

Corresponding authors
Yingli Lian, yllian33@163.com
Baojun Shi, shibj@haid.com.cn

## ABSTRACT

To maintain the health of aquaculture fish, it is critical to understand the composition of microorganisms in aquaculture water and sediment and the factors affecting them. This study examined the water and sediment microbiota compositions of four different types of ponds in South China that were used to culture grass carp (*Ctenopharyngodon idella*) of different sizes through high-throughput sequencing of the 16S rRNA gene, and analyzed their correlations with environmental factors. The results showed that ponds with cultured grass carp of different sizes exhibited significant differences in terms of water physicochemical properties and composition of water and sediment microbiota. Furthermore, the exchange of microorganisms between water and sediment microbiota was lowest in ponds with the smallest grass carp and highest in ponds with the largest grass carp. All detected environmental factors except water temperature were significantly correlated with the water microbiota, and all detected environmental factors in the sediment were correlated with sediment microbiota. Moreover, *Aeromonas hydrophila* and *Vibrio* were significantly increased in the water microbiota, especially in ponds with small juvenile grass carp, implying an increased risk of *A. hydrophila* and *Vibrio* infections in these environments. Our results provide useful information for the management of grass carp aquaculture ponds.

## INTRODUCTION

Aquaculture is an important source of high-quality protein for humans, providing 15–20% of the animal protein consumed by >4 million people worldwide (*Tezzo et al., 2021*). Currently, intensive aquaculture is the main method of aquaculture in China because of its advantages in boosting the output of aquatic products and profits (*Dai & Dong, 2014*;

*Edwards, 2015*; *Zou & Huang, 2015*). However, intensive aquaculture, characterized by high density and high feed loading, can deteriorate aquaculture water quality, especially during the later stages of cultivation (*Dauda et al., 2019*). This not only affects the growth of cultured fish directly (*Colt et al., 1981*; *Alcaraz & Espina, 1997*; *Cheng et al., 2015*; *Wang et al., 2016*), but may also cause excessive growth of conditional pathogens, eventually posing a threat to the health of cultured fish (*Tengs & Rimstad, 2017*; *Li et al., 2019*).

Various microorganisms live in aquaculture ponds and participate in the metabolism of nutrients in aquaculture water and sediments, which play an important role in maintaining water quality (*Ni et al., 2018*; *Li et al., 2020b*). Simultaneously, microorganisms also interact with aquatic life and potentially lead to bacterial diseases in farmed fish (*Zeng et al., 2020*; *Liu et al., 2021*; *Jing et al., 2021*; *Zhang et al., 2022*). To ensure the health of aquaculture fish, it is necessary to clarify the composition of microorganisms in aquaculture water and sediment, and the factors affecting them. Moreover, microorganism exchanges between the pond water and sediment microbiota (*Liu et al., 2021*; *Zheng et al., 2021*) probably affects the distribution of microorganisms in pond systems and the health of aquatic organisms (*Wu et al., 2012*). However, the effects of such exchanges on microbiota metabolism and aquatic organisms have not been widely studied.

Grass carp (*Ctenopharyngodon idella*) is the highest produced species in freshwater aquaculture in China, and Guangdong Province is the highest produced province of grass carp in China, accounting for approximately 15.75% of the national total production (*Fisheries , 2022*). Therefore, in this study, we investigated water and sediment microbiota compositions and their correlation with environmental factors in grass carp culture ponds in Guangdong Province, China. Our findings provide valuable insights that can be used as references for effective aquaculture pond management.

## MATERIALS & METHODS

### Aquaculture ponds and sample collection

Water and sediment samples were collected from grass carp aquaculture ponds in the Nansha district (22.61°N, 113°E), Guangdong Province, China, on May 31, 2018. Four types of ponds (three ponds for each type) cultured with different sizes of grass carp, that is, larval fish (LF), small juvenile fish (SJ), middle juvenile fish (MJ), and large juvenile fish (LJ) were sampled. Each pond is 1.5 km$^2$ in area and approximately 2.0 m in depth without water exchange. Clay was removed from all ponds, and the sediments were further disinfected with quicklime before culturing the grass carp. The ponds used for culture had lasted for nearly one year before sampling. The pond management during the culturing process is consistent with our previous report (*Zhang et al., 2020*). The larval fish had a body weight of approximately 1.0 g and a culture density of 750 individuals per m$^2$. The small juvenile fish weighed approximately 200.0 g of body weight and their culture density was 30 individuals per m$^2$. The middle juvenile fish were approximately 310.0 g in body weight, and their culture density was 10 individuals per m$^2$. The large juvenile fish had a body weight of approximately 580.0 g and a culture density of five individuals per m$^2$. Larval and juvenile fish were fed commercial crumbled and pelleted formulated feeds at a ratio of

5% body weight per day. Three surface waters approximately 50 cm below the water level (approximately 1 L) and three upper (0–8 cm) sediment (approximately 500 g) samples were collected from the left, center, and right of each pond using a 5 L hydrophore sampler and Van Veen Grab sampler, respectively (*Zhang et al., 2020*). All samples were stored in an ice box and brought back to the laboratory. A subset of approximately 50 g of each sediment sample was separated and frozen at −80 °C for further DNA extraction. Water samples (approximately 500 mL) were filtered using glass fiber (GF/C) with a 0.22 μm aperture for extracting microbial genomic DNA (*Chen et al., 2021*). The remaining sediment and water samples were used for physicochemical analysis.

## Physicochemical analysis

Dissolved oxygen (DO), pH, and water temperature were measured *in situ* using a multi-parameter water quality probe YSI EXO2 (Yellow Springs Instruments, Yellow Springs, OH, USA). The transparency of the ponds was measured using a Secchi disk. Ammonia, nitrite, nitrate, total nitrogen (TN), phosphate, and total phosphorus (TP) were measured, as described by *Zheng et al. (2017a)*, *Zheng et al. (2017b)*. Chlorophyll-a (Chla) content was measured using a spectrophotometer and calculated as described by *Lichtenthaler (1987)*. Dissolved organic carbon (DOC) was analyzed using a Formacs total organic carbon analyzer (Skalar, Breda, Netherlands). Particle-organic carbon (POC), total suspended solids (TSS), and total organic carbon (TOC) were calculated according to *Zheng et al. (2021)*.

Sediment physicochemical variables were analyzed according to *Zhang et al. (2020)*. Briefly, 10 g of each sediment sample was separated to measure the TOC, TN, and total sulfur (TS) using a PRIMACS TOC analyzer (Skalar, Breda, Netherlands) and a CHNS/O elemental analyzer (Vario EL cube, Germany) for TNS and TS, respectively. TP in the dried sediment was measured by inductively coupled plasma-atomic emission spectrometry (ICP-AES), as previously described by *Murray, Miller & Kryc (2000)*.

## DNA extraction and high-throughput sequencing of 16S rRNA gene amplicon

Water filtration membranes were cut into small pieces with sterilized scissors before DNA extraction (*Ni et al., 2010*). Subsequently, total microbial DNA in the water and sediment was extracted using a PowerSoil DNA isolation kit (MO BIO Laboratories, Carlsbad, CA, USA) according to the manufacturer's instructions. The concentration and purity of the DNA were determined using a NanoDrop One spectrophotometer (Thermo Fisher Scientific, Waltham, MA, USA). The DNA was diluted to 10 ng/μL and stored at −80 °C for further analyses.

The V4 region of the 16S rRNA gene was amplified using the primer pair 515F (5′-GTGYCAGCMGCCGCGGTAA-3′) and 806R (5′-GGACTACNVGGGTWTCTAAT-3′), as previously described (*Zhang et al., 2020*). The amplified DNA fragments were then high-throughput sequenced on Illumina HiSeq platforms with paired-end sequencing at the Biomarker Technologies Corporation (Beijing, China).

Raw paired-end reads were trimmed using Trimmomatic v.0.36 (*Bolger, Lohse & Usadel, 2014*) to remove "N" bases, adaptor sequences, and bases with *Q*-values <20 to obtain

high-quality fragments. The high-quality fragments were merged using FLASH 1.2.8 software (*Magoc & Salzberg, 2011*) and processed using Quantitative Insights into Microbial Ecology (QIIME) 1.9.1 (*Caporaso et al., 2010*). Briefly, the barcoding and primer sequences were removed from the merged sequences using split_libraries.py command in QIIME 1.9.1. Subsequently, chimeric sequences were identified and removed using the Uchime algorithm (*Edgar et al., 2011*) embedded in QIIME 1.9.1 before operational taxonomic unit (OTUs) clustering. The OTUs were then clustered using USEARCH v11 (*Edgar, 2013*) based on the Ribosomal Database Project (RDP) database 18 (*Maidak et al., 1996*), and an OTU table was generated using the UNOISE2 method with a 97% cut-off (*Edgar, 2016*). The taxonomy of each OTU was annotated using the RDP classifier (*Wang et al., 2007*) embedded in QIIME 1.9.1 based on Silva database release 132.

## Data analysis

Microbial community dissimilarities were visualized by principal component analysis (PCA) using the R vegan package (*Dixon, 2003*). Permutational multivariate analysis of variance (PERMANOVA) was conducted using the R vegan package to detect differences among the microbiota of different groups. Source tracking of the microbiota was conducted using SourceTracker (*Knights et al., 2011*). Spearman's correlation coefficient was used to determine the correlation between environmental factors, and between environmental factors and dominant microorganisms. A correlation between microbial communities and environmental factors was determined using Mantel tests and distance-based redundancy analysis (db-RDA) using the R vegan package (*Borcard, Gillet & Legendre, 2011*). Statistical significance of the RDA model was tested using Monte Carlo permutation tests with 999 permutations. Wilcoxon rank-sum exact test and Kruskal–Wallis rank-sum test with Dunn's *post-hoc* test were conducted using R with the FSA v.0.9.3 package to detect significance of data differences between groups. $P < 0.05$ was considered as statistically significant.

## RESULTS

### Physicochemical indices of water and sediment in ponds cultured with different sizes of grass carp

Differences in water physicochemical indices between ponds cultured with grass carp of different sizes were more pronounced than those in the sediment (Fig. 1). Water temperature in the MJ and LJ ponds significantly increased by 5.26% and 4.53% of those in the SJ ponds (Kruskal–Wallis rank sum test with Dunn's *post-hoc* test, $P < 0.05$; Fig. 1A). DO and pH of pond water in the LJ ponds significantly decreased by 64.13% and 12.41% of those in the LF pond and 46.71% and 6.59% of those in the MJ pond (Kruskal–Wallis rank sum test with Dunn's *post-hoc* test, $P < 0.05$; Figs. 1B and 1C). TOC concentrations in the MJ and LJ waters significantly decreased by 46.31% and 42.87% of those in the LF water, and 31.69% and 27.30% of those in the SJ water (Kruskal–Wallis rank sum test with Dunn's *post-hoc* test, $P < 0.05$; Fig. 1D). Water DOC concentrations in the MJ and LJ ponds significantly decreased by 54.91% and 57.16% of those in the LF ponds (Kruskal–Wallis rank sum test with Dunn's *post-hoc* test, $P < 0.05$; Fig. 1E). The water POC concentrations

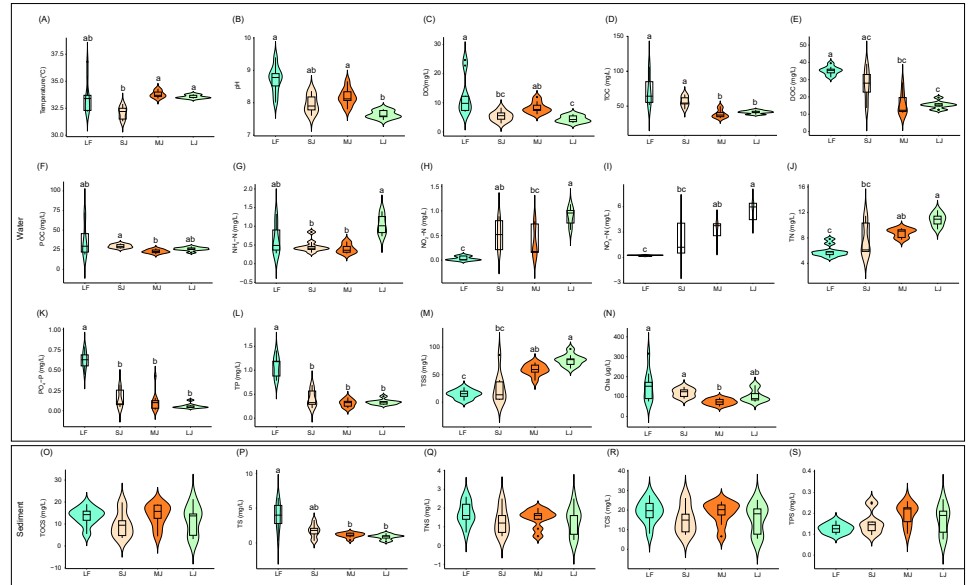

**Figure 1** (A–S) Variations in physical and chemical properties of water and sediment in ponds cultured with different sizes of grass carp. Four types of ponds with three ponds for each type were investigated. Three surface waters approximately 50 cm below the water level (approximately 1 L) and three upper (0–8 cm) sediment (approximately 500 g) samples were collected from the left, center, and right of each pond using a 5 L hydrophore sampler and Van Veen Grab sampler to study, respectively. LF, larval fish; SJ, small juvenile fish; MJ, middle juvenile fish; LJ, large juvenile fish. DO, dissolved oxygen; TOC, total organic carbon; DOC, dissolved organic carbon; POC, particle-organic carbon; TN, total nitrogen; TP, total phosphorus; TSS, total suspended solids; Chla, chlorophyll-a; TOCS, total organic carbon in sediment; TS, total sulfur; TNS, total nitrogen in sediment; TCS, total carbon in sediment; TPS, total phosphorus in sediment. Different letters above the boxplots indicate significant differences.

in the MJ ponds significantly reduced by 22.26% of those in the SJ ponds (Kruskal–Wallis rank sum test with Dunn's *post-hoc* test, $P < 0.05$; Fig. 1F). Water $NH_4^+$-N concentrations in the SJ and MJ ponds significantly reduced by 55.20% and 61.49% of those in the LJ ponds (Kruskal–Wallis rank sum test with Dunn's *post-hoc* test, $P < 0.05$; Fig. 1G). The water $NO_2^-$-N concentrations in the SJ and LJ ponds significantly increased by 15.31 and 26.87 times of those in the LF ponds (Kruskal–Wallis rank sum test with Dunn's *post-hoc* test, $P < 0.05$; Fig. 1H). Water $NO_3^-$-N, TN, and TSS concentrations in the LF ponds significantly reduced by 92.56%, 32.53%, and 75.69% of those in the MJ ponds and 95.57%, 45.47%, and 81.24% of those in the LJ ponds, respectively (Kruskal–Wallis rank sum test with Dunn's *post-hoc* test, $P < 0.05$; Figs. 1I, 1J, and 1M). Water $PO_4^{3-}$-P and TP concentrations in the LF ponds were significantly higher than those in the other kinds of ponds (Kruskal–Wallis rank sum test with Dunn's *post-hoc* test, $P < 0.05$; Fig. 1K and 1L). The water Chla contents in the MJ ponds significantly reduced by 52.13%, and 38.42% of those in the LF and SJ ponds, respectively (Kruskal–Wallis rank sum test with Dunn's *post-hoc* test, $P < 0.05$; Fig. 1N). The above results potentially indicate that the management modes of different kinds of ponds cultured with different sizes of grass carp were different, leading to significant differences in water physicochemical indices. Moreover, the water

$NH_4^+$-N, $NO_2^-$-N, $NO_3^-$-N and TN in the ponds exhibited an increasing trend with the size of the cultured fish, whereas the water $PO_4^{3-}$-P and TP exhibited a decreasing trend.

Among the five physicochemical indices measured in sediment, the TS content in the LF ponds was significantly higher than that in the MJ and LJ ponds (Kruskal–Wallis rank sum test with Dunn's *post-hoc* test, $P < 0.05$; Fig. 1P), and no significant differences were found in terms of the other physicochemical indices (Kruskal–Wallis rank sum test with Dunn's *post-hoc* test, $P \geq 0.05$; Figs. 1O, 1Q–1S). This potentially implies that the pond management mode has a relatively smaller impact on the physicochemical indices of sediment than on those of water. Moreover, the sediment TS exhibited a decreasing trend with the size of the cultured fish.

## Microbiota structure of water and sediment in ponds cultured with different sizes of grass carp

The richness and abundance-based coverage estimator (ACE) indices of sediment microbiota were significantly higher than those of the water microbiota (Kruskal–Wallis rank sum test with Dunn's *post-hoc* test, $P < 0.05$; Figs. 2A and 2C), whereas only the Shannon index of the sediment microbiota of the LF and MJ ponds was significantly higher than that of the water microbiota in the same kind of pond (Kruskal–Wallis rank sum test with Dunn's *post-hoc* test, $P < 0.05$; Fig. 2B). Richness, Shannon, and ACE indices of water or sediment microbiota in different kinds of ponds were not significantly different, although these $\alpha$-diversity indices exhibited an increasing trend with the size of the cultured fish (Kruskal–Wallis rank sum test with Dunn's *post-hoc* test, $P \geq 0.05$; Figs. 2A–2C). Except Shannon index of water microbiota in the LJ, and richness and ACE indices of sediment microbiota in the SJ, there was no significantly difference was detected between microbiome replicates (Table S1). However, db-PCA with PERMANOVA showed that not only did the microbiota compositions differ significantly between water and sediment (PERMANOVA, $F = 74.741$, $n = 36$, $R^2 = 0.516$, $P < 0.001$; Fig. 2D), but also that the microbiota compositions in water (PERMANOVA, $F = 13.03$, $n = 9$, $R^2 = 0.550$, $P < 0.001$) and sediment (PERMANOVA, $F = 14.361$, $n = 9$, $R^2 = 0.574$, $P < 0.001$) were significantly different between different kinds of ponds cultured with different sizes of grass carp (Fig. 2D).

Acidobacteria, Actinobacteria, Bacteroidetes, Chlamydiae, Chloroflexi, Cyanobacteria, Firmicutes, Fusobacteria, Gemmatimonadetes, Patescibacteria, Planctomycetes, Proteobacteria, and Verrucomicrobia dominated the pond water microbiota (Fig. 3A), whereas Euryarchaeota, Nanoarchaeaeota, Acetothemia, Acidobacteria, Actinobacteria, Bacteroidetes, Chloroflexi, Cyanobacteria, Epsilonbacteraeota, Firmicutes, Fusobacteria, Gemmatimonadetes, Kiritimatiellaeota, Nitrospirae, Patescibacteria, Planctomycetes, Proteobacteria, Spirochaetes, and Verrucomicrobia dominated pond sediment microbiota (Fig. 3B). Although significant differences were found in the most dominant phyla in the water among the four types of ponds, only the relative abundances of Chlamydiae (0.085 ± 0.061%, 0.328 ± 0.048%, 0.399 ± 0.216%, and 0.730 ± 0.196% for LF, SJ, MJ, and LJ, respectively), Firmicutes (0.361 ± 0.207%, 0.362 ± 0.087%, 0.565 ± 0.098%, and 0.786 ± 0.070% for LF, SJ, MJ, and LJ, respectively), Fusobacteria (0.086 ± 0.114%, 0.390

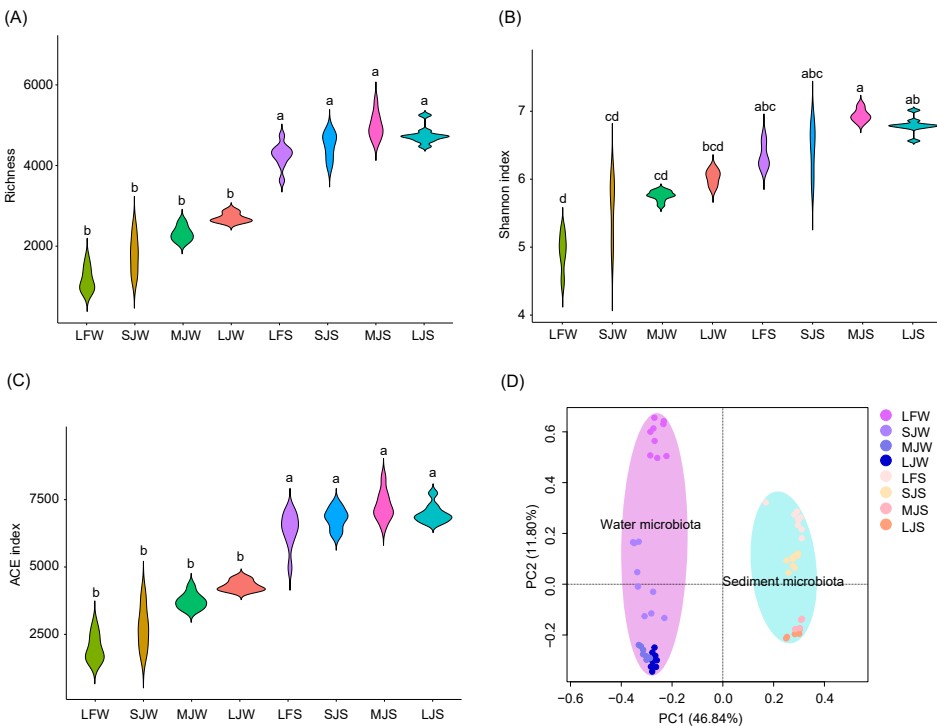

**Figure 2** (A–D) Diversity of water and sediment microbiota in different kinds of ponds cultured with different sizes of grass carp. Four types of ponds with three ponds for each type were investigated. Three surface waters approximately 50 cm below the water level (approximately 1 L) and three upper (0–8 cm) sediment (approximately 500 g) samples were collected from the left, center, and right of each pond using a 5 L hydrophore sampler and Van Veen Grab sampler to study, respectively. LF, larval fish; SJ, small juvenile fish; MJ, middle juvenile fish; LJ, large juvenile fish; W, water; S, sediment. Different lowercase letters above the boxplots indicate significant differences.

± 0.296%, 0.743 ± 0.280%, and 3.250 ± 2.043% for LF, SJ, MJ, and LJ, respectively), and Patescibacteria (0.183 ± 0.090%, 1.428 ± 0.386%, 1.293 ± 0.163%, and 2.017 ± 0.394% for LF, SJ, MJ, and LJ, respectively) increased with the size of cultured grass carp (Fig. 3C). Similarly, the relative abundances of Bacteroidetes (8.171 ± 1.283%, 9.671 ± 1.340%, 11.778 ± 0.379%, and 12.636 ± 0.880% for LF, SJ, MJ, and LJ, respectively), Firmicutes (1.383 ± 0.540%, 2.318 ± 0.518%, 6.153 ± 1.392%, and 4.043 ± 0.389% for LF, SJ, MJ, and LJ, respectively), Fusobacteria (0.158 ± 0.471%, 0.864 ± 0.543%, 1.097 ± 0.485%, 0.899 ± 0.472% for LF, SJ, MJ, and LJ, respectively), and Patescibacteria (0.655 ± 0.142%, 0.867 ± 0.121%, 1.546 ± 0.150%, 1.694 ± 0.112% for LF, SJ, MJ, and LJ, respectively) in sediment increased with the size of the cultured grass carp, whereas the relative abundances of Acidobacteria (3.849 ± 0.529%, 3.849 ± 0.459%, 3.471 ± 0.393%, 2.921 ± 0.347% for LF, SJ, MJ, and LJ, respectively), Kiritimatiellaeota (1.128 ± 0.327%, 0.911 ± 0.139%, 1.020 ± 0.098%, 0.761 ± 0.116% for LF, SJ, MJ, and LJ, respectively), Nitrospirae (1.179 ± 0.373%, 1.270 ± 0.567%, 0.679 ± 0.240%, 0.505 ± 0.144% for LF, SJ, MJ, and LJ, respectively), and Planctomycetes (5.195 ± 1.330%, 4.348 ± 0.589%, 3.288 ± 0.481, 2.927

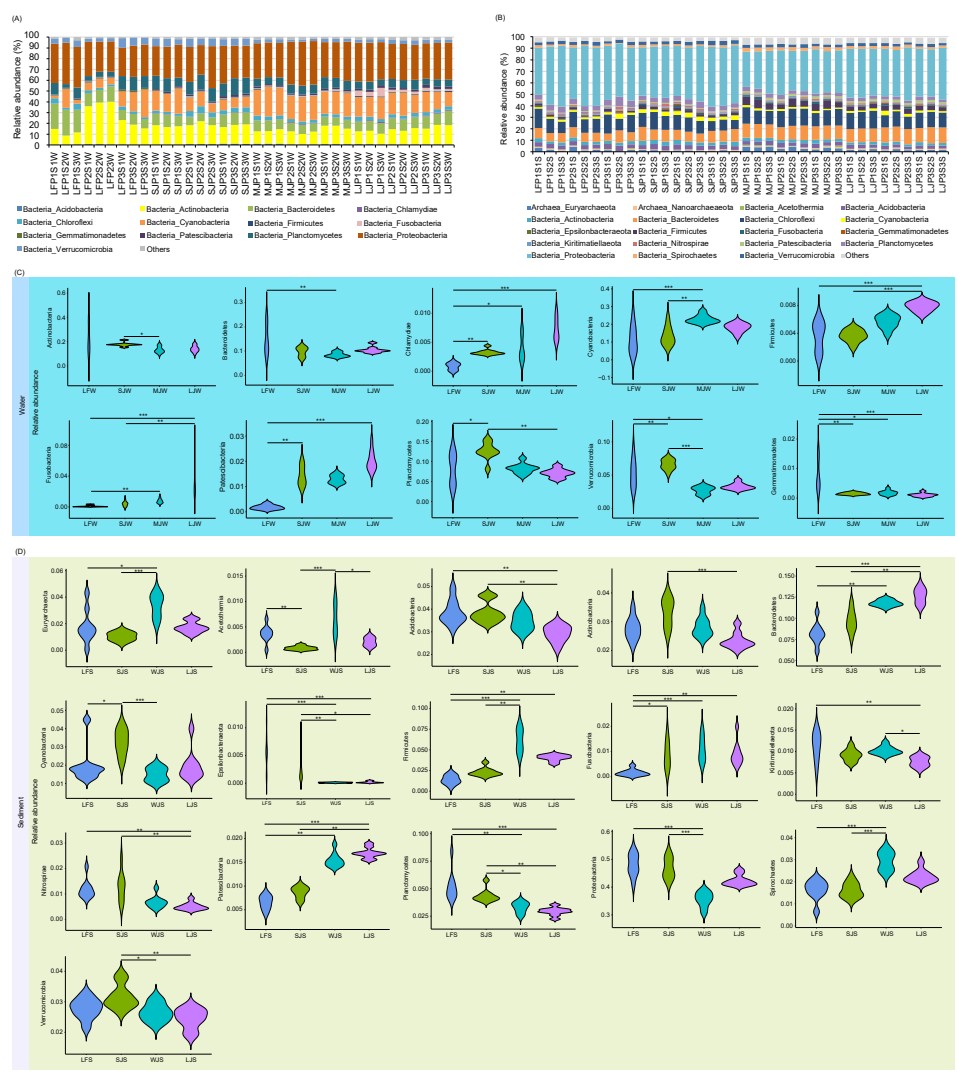

**Figure 3** **(A–D) Composition and differences in dominant phyla in water and sediment microbiota between different ponds cultured with different sizes of grass carp.** Four types of ponds with three ponds for each type were investigated. Three surface waters approximately 50 cm below the water level (approximately 1 L) and three upper (0–8 cm) sediment (approximately 500 g) samples were collected from the left, center, and right of each pond using a 5 L hydrophore sampler and Van Veen Grab sampler to study, respectively. LF, larval fish; SJ, small juvenile fish; MJ, middle juvenile fish; LJ, large juvenile fish. * $P <$ 0.05; ** $P < 0.01$; *** $P < 0.001$.

± 0.367% for LF, SJ, MJ, and LJ, respectively) decreased with the size of the cultured fish (Fig. 3D).

A heatmap with hierarchical clustering based on dominant OTUs showed that water and sediment microbiota were first completely clustered into different groups, and then the water microbiota were completely clustered according to the types of ponds cultured with grass carp of different sizes. Except for the three samples of LJP2S3S, LJP3S3S, and LJP2S1S, the other sediment microbiota were clustered according to the type of pond

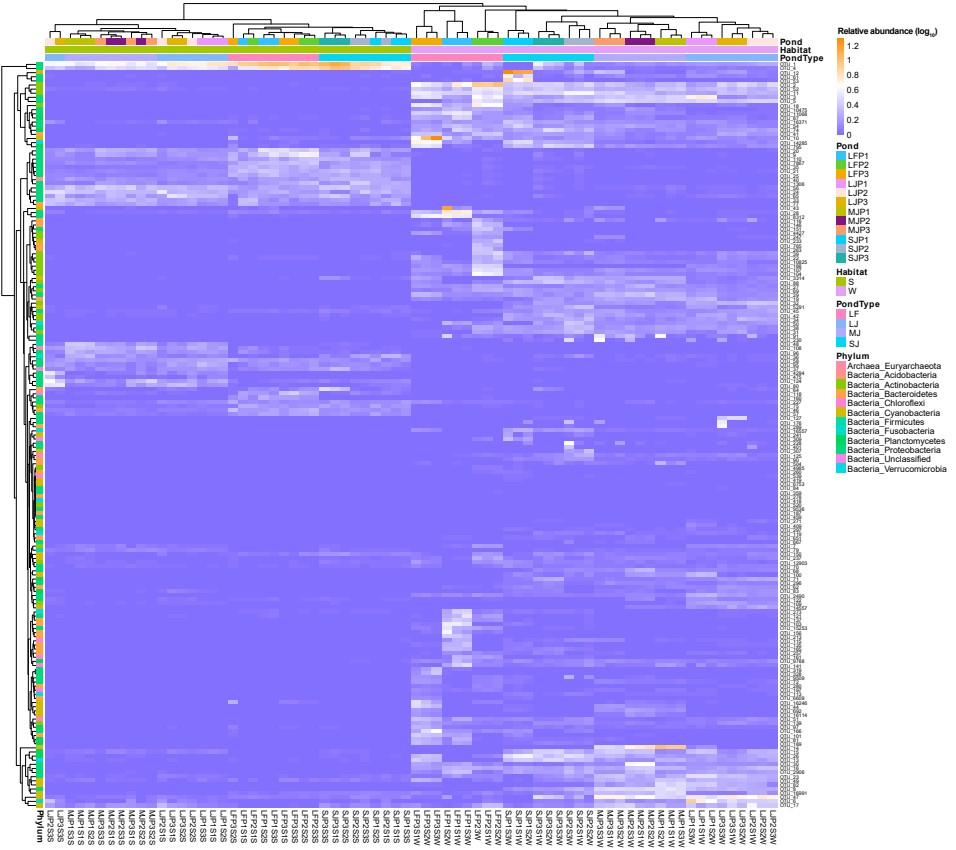

**Figure 4  Heatmap profile shows the composition of dominant OTUs in the water and sediment microbiota in different kinds of ponds cultured with different sizes of grass carp.** LF, larval fish; SJ, small juvenile fish; MJ, middle juvenile fish; LJ, large juvenile fish.

(Fig. 4). Furthermore, water microbiota in different ponds with the same size of grass carp were also completely clustered according to ponds, whereas sediment microbiota were not clustered according to ponds (Fig. 4). These findings suggest that the management practices of ponds stocked with grass carp of varying sizes significantly influenced the dominant OTUs in both water and sediment microbiota, with a stronger impact on water microbiota than on sediment microbiota. The dominant OTUs in the water microbiota differed notably among the ponds, whereas the dominant OTUs in sediment microbiota showed less significant differences across the ponds (Fig. 4).

The linear discriminant analysis of effect size (LEfSe) results showed that many kinds of bacteria in Bacteroidetes, Cyanobacteria, Actinobacteria, Planctomycetes, and Verrucomicrobia were significantly enriched in the water microbiota, whereas many kinds of bacteria in Euryarchaeota, Acidobacteria, Firmicutes, and Fusobacteria were significantly enriched in sediment microbiota (linear discriminant analysis (LDA) score >2; Fig. S1). Different bacteria in Chloroflexi and Proteobacteria were significantly enriched in the water and sediment microbiota (LDA score >2; Fig. S1). In particular, *Methanosaeta* (0.513

± 0.474% and 0.012 ± 0.017% in sediment and water microbiota), *Actibacter* (0.280 ± 0.480% and 0.052 ±0.112%), *Ignavibacterium* (0.404 ± 0.278% and 0.015 ± 0.014%), RBG_16_58_14 (0.289 ± 0.385% and 0.004 ± 0.005%), *Clostridium sensu stricto* 1 (0.154 ±0.123% and 0.077 ± 0.061%), *Hydrogenispora* (0.358 ± 0.459% and 0.008 ± 0.009%), *Cetobacterium* (0.389 ± 0.348% and 0.100 ± 0.109%), *Desulfobacterium* (0.162 ± 0.282% and 0.008 ±0.017%), Sva0081 sediment group (0.616 ± 0.395% and 0.018 ± 0.017%), *Geobacter* (0.425 ± 0.232% and 0.022 ± 0.025%), *Geothermobacter* (1.096 ± 0.378% and 0.069 ± 0.063%), *Limnohabitans* (0.003 ± 0.004% and 0.002 ± 0.002%), *Ottowia* (0.330 ± 0.176% and 0.128 ± 0.099%), *Thiobacillus* (1.076 ±0.520% and 0.038 ± 0.031%), *Dechloromonas* (3.611 ± 2.301% and 0.224 ± 0.260%), *Uliginosibacterium* (0.227 ± 0.496% and 0.007 ±0.009%), and *Candidatus* Competibacter (0.093 ±0.239% and 0.000 ± 0.001%) were significantly enhanced in sediment compared to those in water, whereas the CL500_29 marine group (0.565 ± 0.552% and 0.006 ± 0.004% in water and sediment microbiota), *Mycobacterium* (0.515 ± 0.348% and 0.050 ± 0.037%), *Candidatus* Aquiluna (0.783 ± 1.197% and 0.005 ± 0.007%), *Candidatus* Limnoluna (0.004 ± 0.006% and 0.001 ± 0.002%), *Conexibacter* (0.434 ± 0.430% and 0.040 ±0.015%), *Sediminibacterium* (0.101 ± 0.310% and 0.001 ± 0.001%), *Candidatus* Aquirestis (0.153 ±0.158% and 0.005 ± 0.005%), *Lewinella* (0.159 ±0.321% and 0.004 ± 0.006%), *Phaeodactylibacter* (0.240 ± 0.688% and 0.001 ± 0.003%), *Algoriphagus* (0.074 ± 0.083% and 0.002 ±0.003%), *Fluviicola* (0.141 ± 0.110% and 0.001 ±0.002%), *Wandonia* (0.147 ± 0.529% and 0.001 ±0.002%), *Candidatus* Chloroploca (0.349 ± 0.442% and 0.002 ± 0.004%), *Discoplastis sp*. Banmun 010910B (0.188 ± 0.283% and 0.001 ± 0.003%), *Lepocinclis acus* var. major (0.048 ± 0.080% and 0.000 ± 0.001%), *Lepocinclis* sp. Psurononuma100609I (0.033 ± 0.052% and 0.001 ± 0.002%), *Trachydiscus* (0.877 ± 0.702% and 0.327 ±0.259%), *Microcystis* PCC_7914 (1.395 ± 3.575% and 0.138 ± 0.241%), *Planktothrix* NIVA_CYA15 (1.561 ± 4.287% and 0.006 ± 0.019%), *Nodosilinea* PCC_7104 (0.269 ± 0.328% and 0.001 ± 0.002%), *Cyanobium* PCC_6307 (1.321 ± 2.604% and 0.005 ±0.009%), *Pirellula* (0.465 ± 0.444% and 0.036 ±0.032%), *Rhodopirellula* (0.204 ± 0.329% and 0.016 ± 0.009%), *Roseomonas* (0.301 ± 0.652% and 0.010 ± 0.023%), *Methylocystis* (0.178 ± 0.098% and 0.077 ± 0.028%), *Alsobacter* (0.158 ±0.145% and 0.002 ± 0.003%), *Candidatus* Megaira (0.576 ± 0.309% and 0.009 ± 0.006%), *Novosphingobium* (0.812 ± 0.626% and 0.090 ±0.060%), *Aeromonas hydrophila* subsp. hydrophila (0.363 ± 0.590% and 0.025 ± 0.028%), *Rheinheimera* (0.483 ± 0.821% and 0.001 ± 0.002%), *Hydrogenophaga* (0.548 ± 1.163% and 0.016 ±0.027%), *Kerstersia* (0.006 ± 0.018% and 0.000 ±0.000%), *Limnobacter* (1.397 ± 0.691% and 0.002 ± 0.002%), MWH_Uni P1 aquatic group (0.941 ± 0.311% and 0.002 ± 0.004%), *Massilia* (0.070 ± 0.264% and 0.000 ± 0.001%), *Polaromonas* (0.041 ±0.064% and 0.000 ± 0.001%), *Polynucleobacter* (1.604 ± 1.208% and 0.008 ± 0.006%), *Candidatus* Methylopumilus (0.472 ± 0.247% and 0.002 ± 0.002%), *Pseudomonas* (0.268 ± 0.710% and 0.003 ± 0.004%), *Vibrio* (0.462 ± 1.461% and 0.003 ± 0.007%), and *Silanimonas* (0.086 ±0.300% and 0.002 ± 0.005%) were significantly enhanced in the water microbiota compared to sediment microbiota (LDA score > 2; Fig. S1 and Table S2).

In water microbiota, CL500_29 marine group, *Mycobacterium*, *Candidatus* Aquiluna, *Candidatus* Limnoluna, *Conexibacter*, *Sediminibacterium*, *Lewinella*, *Wandonia*,

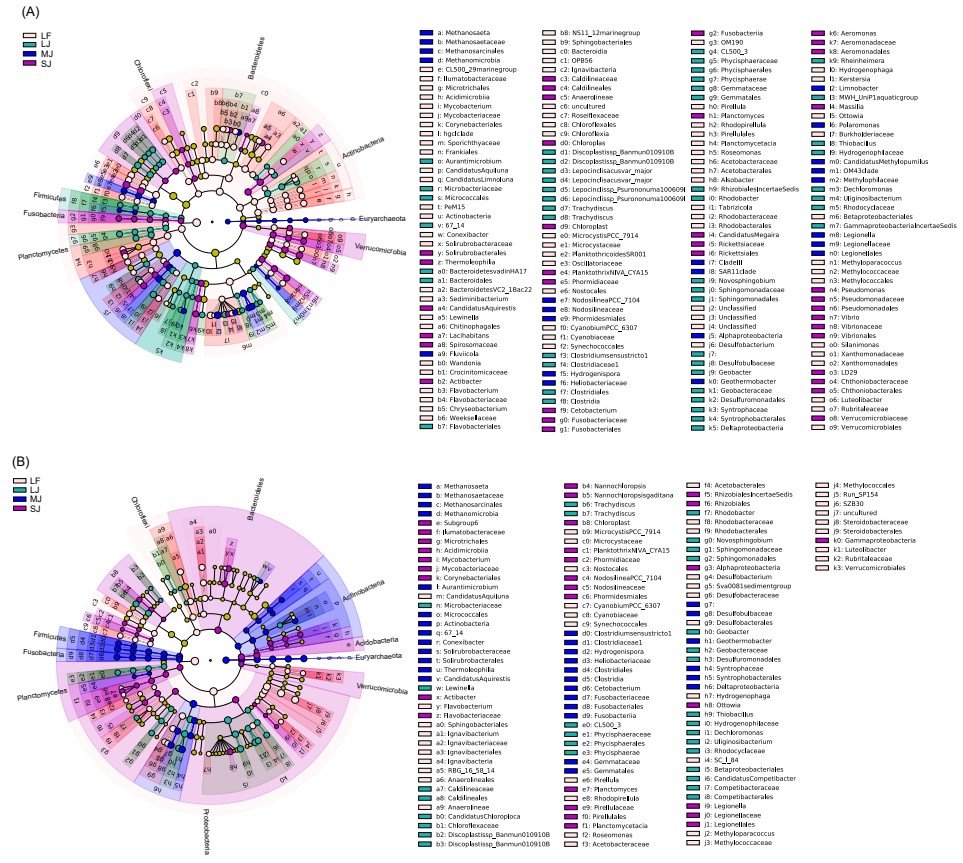

**Figure 5 Cladogram profiles showing LEfSe results for water (A) and sediment (B) microbiota in four kinds of ponds cultured with different sizes of grass carp.** Four types of ponds with three ponds for each type were investigated. Three surface waters approximately 50 cm below the water level (approximately 1 L) and three upper (0–8 cm) sediment (approximately 500 g) samples were collected from the left, center, and right of each pond using a 5 L hydrophore sampler and Van Veen Grab sampler to study, respectively. LF, larval fish; SJ, small juvenile fish; MJ, middle juvenile fish; LJ, large juvenile fish.

*Flavobacterium*, *Chryseobacterium*, *Microcystis* PCC_7914, *Planktothricoides* SR001, *Cyanobium* PCC_6307, *Pirellula*, *Rhodopirellula*, *Roseomonas*, *Alsobacter*, *Tabrizicola*, *Desulfobacterium*, *Hydrogenophaga*, *Kerstersia*, *Ottowia*, *Methyloparacoccus*, *Silanimonas*, and *Luteolibacter* were enhanced in the LF ponds; *Candidatus* Aquirestis, *Lacihabitans*, *Actibacter*, *Planktothrix* NIVA_CYA15, *Cetobacterium*, *Planctomyces*, *Candidatus* Megaira, *A. hydrophila* subsp. hydrophila, *Massilia*, *Pseudomonas*, and *Vibrio* were enhanced in the SJ ponds; *Methanosaeta*, *Fluviicola*, *Nodosilinea* PCC_7104, *Geothermobacter*, *Limnobacter*, *Polaromonas*, *Candidatus* Methylopumilus, and *Legionella* were enhanced in the MJ ponds; and *Aurantimicrobium*, *Discoplastis* sp. *Banmun* 010910B, *Lepocinclisacus* var. major, *Lepocinclis* sp. *Psurononuma* 100609I, Clostridium sensu stricto 1, *Rhodobacter*, *Novosphingobium*, *Geobacter*, *Rheinheimera*, MWH_UniP1 aquatic group, *Thiobacillus*, *Dechloromonas*, and *Uliginosibacterium* were enhanced in LJ ponds (LDA score > 2; Fig. 5A).

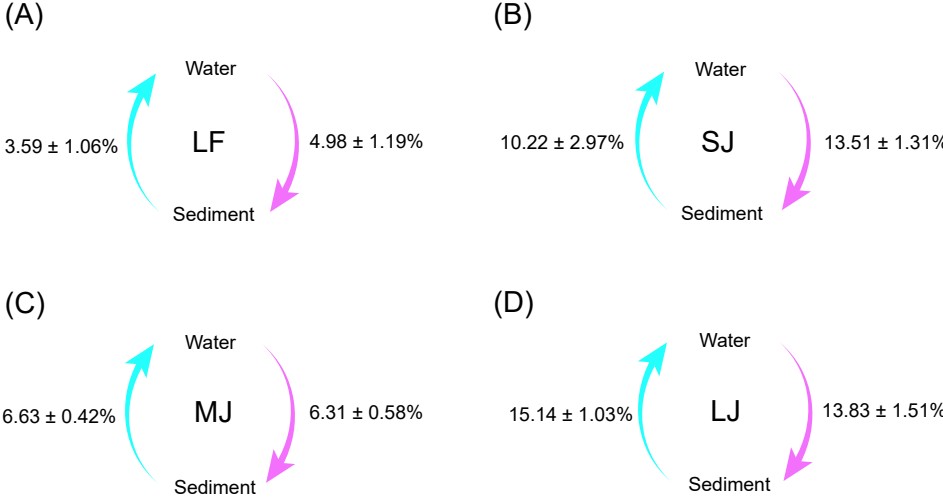

**Figure 6** **(A–D) Source tracking analysis showing the proportions of microorganisms from sediment to water microbiota, and from water to sediment microbiota in different kinds of ponds cultured with different sizes of grass carp.** Four types of ponds with three ponds for each type were investigated. Three surface waters approximately 50 cm below the water level (approximately 1 L) and three upper (0–8 cm) sediment (approximately 500 g) samples were collected from the left, center, and right of each pond using a 5 L hydrophore sampler and Van Veen Grab sampler to study, respectively. LF, larval fish; SJ, small juvenile fish; MJ, middle juvenile fish; LJ, large juvenile fish.

In sediment microbiota, *Candidatus Aquiluna*, *Flavobacterium*, *Ignavibacterium*, R8G_16_58_14, *Microcystis* PCC_7914, *Cyanobium* PCC_6307, *Pirellula*, *Rhodopirellula*, *Roseomonas*, *Desulfobacterium*, Sva0081 sediment group, *Hydrogenophaga*, *Methyloparacoccus*, and *Luteolibacter* were enhanced in the LF ponds; *Mycobacterium*, *Actibacter*, *Nannochloropsis*, *Planktothrix* NIVA_CYA15, *Nodosilinea* PCC_7104, *Planctomyces*, *Ottowia*, and *Legionella* were enhanced in the SJ ponds; *Methanosaeta*, *Aurantimicrobium*, *Conexibacter*, *Candidatus* Aquirestis, *Clostridium* sensu stricto 1, *Hydrogenispora*, *Cetobacterium*, and *Geothermobacter* were enhanced in the MJ ponds; and *Lewinella*, *Candidatus* Chloroploca, *Discoplastis* sp. Banmun010910B, *Trachydiscus*, *Rhodobacter*, *Novosphingobium*, *Geobacter*, *Thiobacillus*, *Dechloromonas*, *Uliginosibacterium*, and *Candidatus* Competibacter were enhanced in LJ ponds (LDA score >2; Fig. 5B).

Source tracking results showed that the exchange proportions of microorganisms in the water and sediment microbiota were lowest in the LF ponds and highest in the LJ ponds (Fig. 6). Simultaneously, there was no significant difference between the proportions of microorganisms from sediment to water microbiota and the proportions of microorganisms from water to sediment microbiota in all kinds of ponds (Wilcoxon rank sum exact test, $P$ < 0.05; Fig. 6).

## Correlation between microbiota composition and physicochemical indices in water and sediment

RDA with Monte Carlo test results showed that all water physicochemical indices except temperature were significantly correlated with water microbiota (Monte Carlo test, $P$

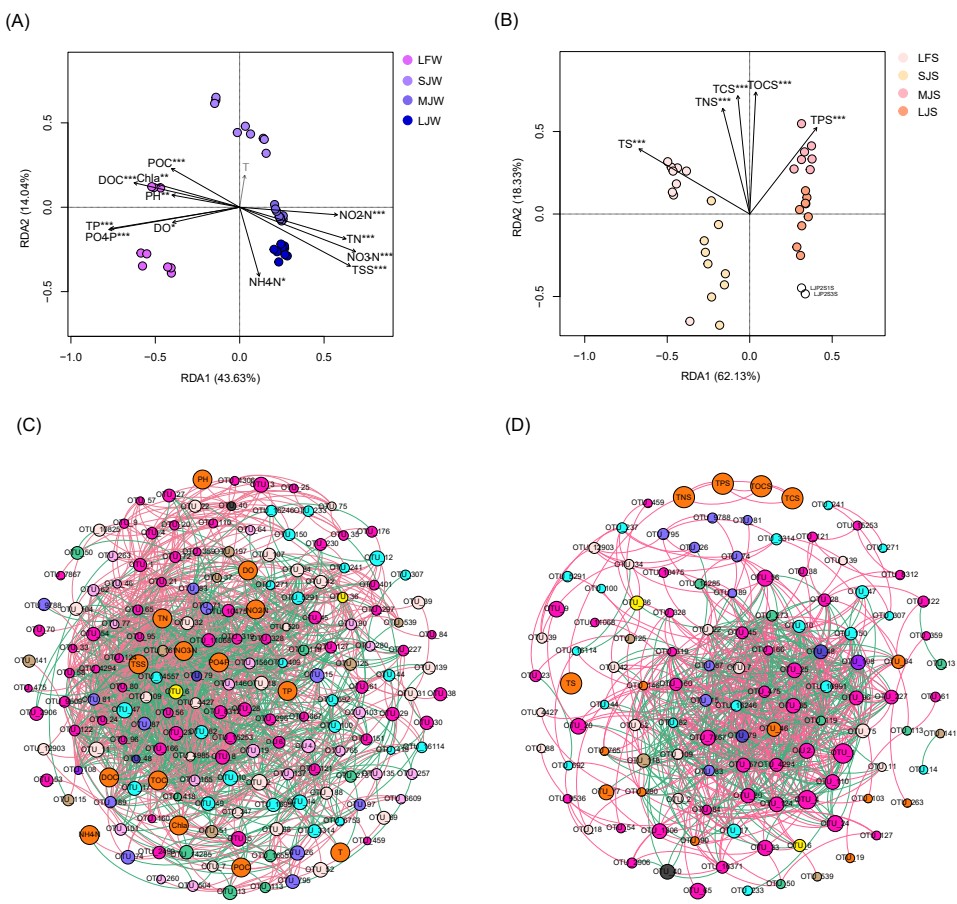

**Figure 7** **RDA profiles (A and B) and co-occurrence networks (C and D) displaying correlation between pond environmental factors and water (A and C) and sediment (B and D) microbiota in different kinds of ponds cultured with different sizes of grass carp.** Four types of ponds with three ponds for each type were investigated. Three surface waters approximately 50 cm below the water level (approximately 1 L) and three upper (0–8 cm) sediment (approximately 500 g) samples were collected from the left, center, and right of each pond using a 5 L hydrophore sampler and Van Veen Grab sampler to study, respectively. LF, larval fish; SJ, small juvenile fish; MJ, middle juvenile fish; LJ, large juvenile fish. T, water temperature; DO, dissolved oxygen; TOC, total organic carbon; DOC, dissolved organic carbon; POC, particle-organic carbon; TN, total nitrogen; TP, total phosphorus; TSS, total suspended solids; Chla, chlorophyll-a; TOCS, total organic carbon in sediment; TS, total sulfur; TNS, total nitrogen in sediment; TCS, total carbon in sediment; TPS, total phosphorus in sediment.

< 0.05; Fig. 7A), and all sediment physicochemical indices were significantly correlated with sediment microbiota (Monte Carlo test, $P < 0.05$; Fig. 7B). Co-occurrence network analysis based on the Spearman correlation coefficients of the dominant OTUs and physicochemical indices showed that water physicochemical indices were more significantly correlated with dominant OTUs than sediment physicochemical indices (Spearman correlation coefficient >0.6 and $P < 0.05$; Fig. 7C). There was no significant correlation between sediment physicochemical indices and dominant OTUs (Fig. 7D), suggesting that these physicochemical indices may affect the microbiota structure through non-dominant OTUs.

Spearman correlation analysis also showed that water $NH_4^+$-N concentration was significantly positively correlated with the relative abundances of *Aurantimicrobium* sp., *Candidatus* Aquiluna sp., *Candidatus* Competibacter sp., *Cetobacterium* sp., *Chryseobacterium* sp., *Conexibacter* sp., *Dechloromonas* sp., *Discoplastis* sp. Banmun010910B, *Ignavibacterium* sp., *Lepocinclis* sp. Psurononuma 100609I, *Limnobacter* sp., *Mycobacterium* sp., *Ottowia* sp., *Rhodobacter* sp., *Sediminibacterium* sp., *Trachydiscus minutus*, and *Uliginosibacterium* sp., and significantly negatively correlated with the relative abundances of *Cyanobium* PCC-6307, *Fluviicola* sp., *Pirellula* sp., *Planktothricoides* sp. SR001, and *Rhodopirellula* sp. ($P < 0.05$; Fig. S2A). Water $NO_2^-$-N concentration was significantly positively correlated with the relative abundances of *Candidatus* Aquirestis sp., *Candidatus* Competibacter sp., *Cetobacterium* sp., *Clostridium perfringens*, *Cyanobium* PCC-6307, *Dechloromonas* sp., *Discoplastis* sp. Banmun010910B, *Geobacter* sp., *Geothermobacter* sp., *Ignavibacterium* sp., *Lacihabitans* sp., *Legionella* sp., *Lepocinclis acus* var. major, *Lepocinclis* sp. Psurononuma 100609I, *Massilia* sp., *Mycobacterium* sp., *Novosphingobium* sp., *Phacus* sp., *Planktothrix* NIVA-CYA_15, *Pseudomonas* sp., *Rheinheimera* sp., *Rhodobacter* sp., *Thiobacillus* sp., *Trachydiscus minutus*, and *Uliginosibacterium* sp., and significantly negatively correlated with the relative abundances of *Algoriphagus* sp., *Candidatus* Aquiluna sp., *Candidatus* Limnoluna sp., *Candidatus* Methylopumilus sp., *Chryseobacterium* sp., *Cyanobium* PCC-6307, *Desulfobacterium* sp., *Flavobacterium* sp., *Hydrogenophaga* sp., *Kerstersia* sp., *Microcystis aeruginosa* DIANCHI905, Ottowia sp., *Pirellula* sp., *Polynucleobacter* sp., *Rhodobacter* sp., *Rhodopirellula* sp., *Roseiflexus* sp., *Roseomonas* sp., *Silanimonas* sp., *Sunechococcus* sp., *Tabrizicola* sp., and *Vibrio cholerae* ($P < 0.05$; Fig. S2A). The water microorganisms that were significantly related to water $NO_3^-$-N and TN were similar. Water $NO_3^-$-N and TN concentrations were significantly positively correlated with the relative abundances of *Candidatus* Aquirestis sp., *Candidatus* Competibacter sp., *Cetobacterium* sp., *Clostridium perfringens*, *Cyanobium* PCC-6307, *Dechloromonas* sp., *Discoplastis* sp. Banmun010910B, *Geobacter* sp., *Geothermobacter* sp., *Ignavibacterium* sp., *Lacihabitans* sp., *Legionella* sp., *Lepocinclis acus* var. major, *Lepocinclis* sp. Psurononuma 100609I, *Massilia* sp., *Methanosaeta* sp., *Mycobacterium* sp., *Novosphingobium* sp., *Phacus* sp., *Rhodobacter* sp., *Thiobacillus* sp., *Trachydiscus* minutus, and *Uliginosibacterium* sp., and significantly negatively correlated with the relative abundances of *Candidatus* Aquiluna sp., *Candidatus* Limnoluna sp., *Candidatus* Methylopumilus sp., *Chryseobacterium* sp., *Cyanobium* PCC-6307, *Desulfobacterium* sp., *Flavobacterium* sp., *Hydrogenophaga* sp., *Kerstersia* sp., *Methyloparacoccus* sp., *Microcystis aeruginosa* DIANCHI905, *Mycobacterium* sp., *Pirellula* sp., *Rhodobacter* sp., *Rhodopirellula* sp., *Roseiflexus* sp., *Roseomonas* sp., *Silanimonas* sp., and *Tabrizicola* sp. ($P < 0.05$; Fig. S2A). The correlation between water $PO_4^{3-}$-P and TP concentration and water microorganisms exhibited an opposite trend to the correlation between water $NO_3^-$-N and TN concentrations and water microorganisms (Fig. S2A). The correlation between Chla, DOC, TOC, and water microorganisms was more similar to that of $PO_4^{3-}$-P and TP compared with those of $NO_3^-$-N and TN (Fig. S2A). Fewer water-dominant microorganisms were significantly correlated with water temperature, which is probably caused by small differences in water temperature (Fig. S2A). Less sediment-dominant microorganisms were significantly related to sediment

physicochemical indices comparing with water-dominant microorganisms (Fig. S2B). These findings suggest that water microbiota in grass carp ponds is more susceptible to environmental parameters than sediment microbiota.

## DISCUSSION

Biodiversity is the basis of ecosystem structure and functional maintenance (*Tilman & Downing, 1994*; *Hulot et al., 2000*). Although the role of microorganisms in nutrient metabolism and circulation in aquaculture ponds has been widely confirmed (*Ni et al., 2018*; *Thurlow et al., 2019*; *Gong et al., 2021*), the composition and function of different types of pond microbial communities and their influencing factors have also been extensively investigated (*Moriarty, 1997*; *Mao et al., 2022*), the impact of different types of ponds cultured with different sizes of fish on pond water and sediment microbiota has not been fully elucidated. Our results indicated that water physicochemical indices of ponds cultured with different sizes of grass carp were more susceptible to the influence of the size of the fish than the sediment physicochemical indices, and the structures of the water and sediment microbiota were also different because of the size of grass carp. This is probably due to the different cultivation and pond management modes of grass carp culture at different stages, which changes the physicochemical indices of the pond and affects the water and sediment microbiota. Moreover, fish microbiota is one of the primary sources of sediment microbiota, and >15% of the sediment microbiota is derived from fish (*Zhang et al., 2022*). Therefore, the differences in water and sediment microbiota in different ponds were probably caused by differences in the gut microbiota of grass carp of different sizes.

Habitat microbiota is an important factor affecting the microbiota structure of aquaculture organisms (*Liu et al., 2021*; *Zhang et al., 2022*; *Giatsis et al., 2015*), and is closely related to the health of aquaculture organisms (*De Schryver et al., 2012*; *Chen et al., 2017*; *Kaktcham et al., 2017*; *Huang et al., 2018*)). *A. hydrophila* (*Song et al., 2017*) , *Citrobacter* spp. (*Lü et al., 2011*), *Aeromonas veronii* (*He et al., 2018*), *Aeromonas sobria* (*Zou et al., 2019*), *Aeromonas allosaccharophila* (*Zou et al., 2019*), *Aeromonas punctata* (*Xu et al., 1987*), *Plesiomonas shigelloides* (*Zou et al., 2019*), *Lactobacillus gasseri* (*Zou et al., 2019*), *Fiexibacter coiumnaris* (*Xu, Wei & Zhang, 2007*), *Vibrio mimicus* (*Li et al., 2020a*), *Vibrio vulnificus* (*Liu et al., 2019*), and *Myxococcus piscicola* (*Lu, Ni & Ge, 1975*; *Huang et al., 1983*) are commonly reported as bacterial pathogens of grass carp. Our results showed that *A. hydrophila* subsp. hydrophila and *Vibrio* were significantly enhanced in terms of water microbiota, especially in the SJ water microbiota. This result implied that grass carp are at an increased risk of infection by *A. hydrophila* and *Vibrio* in SJ ponds.

The microbiota plays a crucial role in facilitating the conversion of various forms of nitrogen and phosphorus in pond ecosystems (*Ni et al., 2018*; *Gong et al., 2021*; *Wang et al., 2022*). Members of *Dechloromonas* were determined as denitrifying polyphosphate-accumulating organisms (*Dai et al., 2017*). *Limnohabitans* spp. participate in nitrogen and phosphorus metabolism as photoautotrophs and ammonia oxidizers (*Zeng et al., 2012*). *Candidatus* Aquiluna, a photoheterotroph (*Kang et al., 2012*), was reportedly positively correlated with the microbial metabolic activity of organic nitrogen (*Lukwambea et al.,*

*2020*). *Pseudomonas furukawaii* ZS1, *Acidovorax facilis*, *Citrobacter diversus*, and certain *Thauera* species also participate in nitrogen removal in aquaculture ponds (*Mai et al., 2021*; *Niu et al., 2022*). Our results showed that *Dechloromonas*, *Limnohabitans*, *Candidatus* Aquiluna, and *Pseudomonas* were detected in the pond water and sediment microbiota. In addition, *Dechloromonas* sp. OTU was significantly positively correlated with water $NH_4^+$-N, $NO_2^-$-N, $NO_3^-$-N, and TN, and was enhanced in LJ pond water and sediment; and *Candidatus* Aquiluna OTU was significantly negatively correlated with water $NO_2^-$-N, $NO_3^-$-N, and TN, and significantly positively correlated with water $NH_4^+$-N (Fig. S2), and was enhanced in LF pond water and sediment. Moreover, our results indicated that water $NO_2^-$-N, $NO_3^-$-N, and TN concentrations were significantly positively correlated with the relative abundance of *Cyanobium* PCC-6307 OTUs, whereas $NH_4^+$-N concentration was significantly negatively correlated, implying that these OTUs may play important roles in nitrogen metabolism of pond water, although further verification is needed. Moreover, *Cyanobium* PCC-6307 was enhanced in LF pond water and sediment. These results implied that different bacterial species participate in nitrogen metabolism in the ponds cultured with grass carp of different sizes. Moreover, the enhancement of *Cyanobium* PCC-6307 in LF pond implied that LF pond was more prone to cyanobacteria bloom.

The impact of environmental factors on the aquatic microbiota community structure has been extensively studied, and water temperature, DO, pH, and nutrients have been found to significantly affect microbiota structure (*Ni et al., 2018*; *Guan et al., 2019*). However, differences in water temperature between the ponds in this study were probably be caused by the different sampling times. The fluctuation of water physicochemical factors during the day and night is greater than that of sediment factors, which led to the sediment microbiota being more stable than water microbiota, consistent with previous research (*Zheng et al., 2021*).

It should be noted that the microbiota structure in aquaculture pond varies seasonally (*Duarte et al., 2019*; *Marmen et al., 2021*), whereas we only collected samples in May and did not track the changes in microbiota structure throughout the entire grass carp aquaculture cycle. Moreover, despite restriction on the use of antibiotics in aquaculture, some antibiotics, such as florfenicol, enrofloxacin, and oxytetracycline, are permitted and extensively used in the aquaculture industry (*Sáenz et al., 2019*; *Miranda, Godoy & Lee, 2018*; *Hossain et al., 2022*), and the distribution of antibiotic resistance profiles in aquaculture water and sediment has received widespread attention (*Mao et al., 2019*; *Wang et al., 2021*; *Zhou et al., 2021*). Although our purpose in this study was to describe the impact of grass carp size on the microbiota structure of aquaculture pond water and sediment, we only used high-throughput sequencing of prokaryotic 16S rRNA gene to study the microbiota structure. Further analysis of antibiotics on microbiota, and antibiotic resistance profiles in aquaculture water and sediment using metagenomic sequencing is needed in the future.

## CONCLUSIONS

Differences in water physicochemical indices between ponds cultured with grass carp of different sizes were more pronounced than those in the sediment. The water $NH_4^+$-N,

$NO_2^--N$, $NO_3^--N$ and TN in the ponds exhibited an increasing trend with the size of the cultured fish, whereas the water $PO_4^{3-}-P$ and TP exhibited a decreasing trend. The pond management mode has a relatively smaller impact on the physicochemical indices of sediment than on those of water. Moreover, the sediment TS exhibited a decreasing trend with the size of the cultured fish. Different types of ponds cultured with grass carp of different sizes exhibited significant differences in composition of the water and sediment microbiota. The differences in water and sediment microbiota in different ponds were probably caused by differences in the gut microbiota of grass carp of different sizes. The exchange of microorganisms between the water and sediment microbiota was lowest in ponds cultured the LF of grass carp and highest in ponds cultured the LJ of grass carp. *A. hydrophila* and *Vibrio* were significantly increased in the water microbiota, especially in ponds cultured the SJ of grass carp. Additionally, there were significant correlations between water parameters including POC, DOC, Chla, pH, TP, $PO_4^{3-}-P$, DO, $NH_4^+-N$, $NO_2^--N$, $NO_3^--N$, TN, and TSS, and the water microbiota. All detected sediment parameters including TS, TNS, TCS, TOCS, and TPS showed correlations with the sediment microbiome.

## ACKNOWLEDGEMENTS

We would like to thank Guangzhou Chengyi Aquatic Technology Co., Ltd., China for providing the experimental field and assisting in conducting the experiment. We would like to thank Jiajia Ni at Guangdong Meilikang Bio-Science Ltd., China for assistance with data analysis and manuscript revision.

### Funding

This research was funded by the 2020 Industrial Talent Policy Project Innovation leading team of Panyu District (No. 2021-R01-4), the Zhejiang Major Program of Science and Technology (No. 2022SNJF063 and 2022C02027), the Natural Science Foundation of Ningbo (No. 2022J050), and the Key Research and Development Program of Ningbo (No. 2022Z172 and 2022Z059). The funders had no role in study design, data collection and analysis, decision to publish, or preparation of the manuscript.

### Grant Disclosures

The following grant information was disclosed by the authors:
2020 Industrial Talent Policy Project Innovation: 2021-R01-4.
Zhejiang Major Program of Science and Technology: 2022SNJF063, 2022C02027.
Natural Science Foundation of Ningbo: 2022J050.
Key Research and Development Program of Ningbo: 2022Z172, 2022Z059.

### Competing Interests

Yingli Lian, Jian Wang, Jiayi Tang, Xuan Zhu and Baojun Shi are employed by Guangdong Haid Group Co., Ltd. Sampling was conducted at a site associated with Guangzhou Chengyi Aquatic Technology Co., Ltd., China, which is partner to one of our affiliations.

## Author Contributions

- Yingli Lian conceived and designed the experiments, performed the experiments, analyzed the data, prepared figures and/or tables, authored or reviewed drafts of the article, and approved the final draft.
- Xiafei Zheng conceived and designed the experiments, performed the experiments, analyzed the data, prepared figures and/or tables, authored or reviewed drafts of the article, and approved the final draft.
- Shouqi Xie conceived and designed the experiments, authored or reviewed drafts of the article, and approved the final draft.
- Dan A performed the experiments, prepared figures and/or tables, and approved the final draft.
- Jian Wang performed the experiments, prepared figures and/or tables, authored or reviewed drafts of the article, and approved the final draft.
- Jiayi Tang performed the experiments, prepared figures and/or tables, and approved the final draft.
- Xuan Zhu performed the experiments, prepared figures and/or tables, and approved the final draft.
- Baojun Shi conceived and designed the experiments, performed the experiments, prepared figures and/or tables, authored or reviewed drafts of the article, and approved the final draft.

## Field Study Permissions

The following information was supplied relating to field study approvals (i.e., approving body and any reference numbers):

Field experiments were approved verbally by Guangzhou Chengyi Aquatic Technology Co., Ltd., China.

## Data Availability

Sequence data are available at GenBank: PRJNA977952

## Supplemental Information

Supplemental information for this article can be found online at http://dx.doi.org/10.7717/peerj.15892#supplemental-information.

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
