# Peer review of "Microbiota composition and correlations with environmental factors in grass carp (Ctenopharyngodon idella) culture ponds in South China"

_PeerJ, doi:10.7717/peerj.15892_

## Round 0.1 · original submission · Major Revisions

Dear Authors, please see the reviewers' comments below. Based on these comments, the manuscript needs to be revised substantially and re-submitted to the journal for consideration.

·

Basic reporting

no comment

Experimental design

no comments

Validity of the findings

Results have been described in detail and explained well in relevance to the figures. There were a few instances noted where the figure or text could be edited for the article to be more scientifically robust.

Additional comments

The article “Microbiota composition and correlations with environmental factors in grass carp culture
ponds in South China” by Lian et al., investigated the bacterial community composition in water and
sediments associated with grass carp culture in ponds. They have used 16S rRNA gene sequencing to
analyze the microbiome and correlated that with measured water and sediment physical-chemical
parameters. Although there are several studies looking at pond microbiota, there are lesser studies
associated with ponds that culture grass carp (https://doi.org/10.1016/j.aqrep.2023.101556). Therefore, the findings are considerably interesting. The paper is well written and easy to follow. This is a study that could potentially add significantly to current knowledge about microbial community structure in aquaculture ponds.
Few points to consider for building scientific rigor:
- Experimental design was well presented with details about the pond size and fish size. Why did
the authors choose this particular aquaculture pond?
- The total number of samples and for each sample type needs to be added. The only figure which
provides some information about sample type is PCA plot in figure 2D. Please add the amount of
variation in microbial community composition explained by each variable in this figure.
- Since the Shannon diversity of sample SJW is variable, it would be good to compare between
microbiome replicates. Was there a difference between biological replicates?
- Bacterial microbiome varies significantly with seasons. Since the authors only sampled once, it
would good to include this point as a limitation of the study.
- Another limitation of the study is the use of 16S rRNA sequencing, that only provides
information about the microbial players present, unlike metagenomic sequencing which can also
provide antibiotic resistance profiles.
Line 357- 358: Are the authors able to provide any explanation as to why the Aeromonas and Vibrio
abundance was higher in the SJW compared to the others? For example, Aeromonas hydrophila is
abundant in ponds and is not considered a pathogen. While it is important to mention that the study
found these organisms, the supplemental figure S1 does not show species level assignment and hence it is not appropriate to report that the results implied “increased risk of infection” (line 358-359).

·

Basic reporting

At first, I would to mention the actual topic of the study – to compare the microbial community and impact of environmental factors to its composition.
Bioinformatics approaches described in the manuscript can be improved. There are strong reasons to use amplicon sequence variants (ASVs) instead of OTUs for less noise and avoid spurious taxons (https://www.mdpi.com/2306-5354/9/4/146).
Callahan, McMurdie and Holmes points that “the improvements in reusability, reproducibility and comprehensiveness are sufficiently great that ASVs should replace OTUs as the standard unit of marker-gene analysis and reporting” Callahan, B., McMurdie, P. & Holmes, S. Exact sequence variants should replace operational taxonomic units in marker-gene data analysis. ISME J 11, 2639–2643 (2017). https://doi.org/10.1038/ismej.2017.119
There are no raw data mentioned in the manuscript, so it’s impossible to reproduce the whole analysis or perform the further meta-analysis. I strongly suggest to add the reference to raw reads (PRJNA579535) to the manuscript, noting that this data have been published in 2020. The previous paper contains the invalid accession numbers (SUB6464626 (16S rRNA), SUB6467726 (dsrB) and SUB6467851 (soxB) – in fact there are “submission” numbers, not public accession numbers)
L66: What is known to date about grass carp microbial community in particular?

Experimental design

L81: how the growing fish can fit the ponds of same size?
Some details of the study must be clarified to better understanding and reproducibility:
L127: how data analysis was performed in details? please provide the reference to the protocol if available.
L131: which version of RDP database was used?
L139: Pearson correlation can be used only for linear relations, how do you hypothesize that?
L142: Mantel test commonly used for comparison phylogenetic trees or distance matrices. How in details you calculated the correlation coefficients?
As said above, using OTUs approach produces a lot of spurious OTUs. L113 shows thousands of OTUs, most of which, I propose, share less than 1%. Such classification is hard to interpret. For further study see, f.e.:
https://doi.org/10.1371/journal.pone.0227434
https://doi.org/10.1038/s43705-021-00033-z

Validity of the findings

The description of results needs to be improved. L155-L176 and L231-270 written by the same template, plain re-telling the figures.
There are multiple phrases in Results like “… were significantly lower/higher than …”, neglecting the evaluation of the effect.
The Conclusion section is written vaguely. It’s interesting to compare the present study results with the similar studies, f.e.: https://doi.org/10.1016/j.scitotenv.2020.142840, https://doi.org/10.1007/s11274-014-1677-1 and https://doi.org/10.1016/j.aquaculture.2023.739421
My notes about the main text:
• L45: remove “Add your introduction here’
• References list has missing idents

---

## Round 0.2 · Minor Revisions

Dear Authors, please refer to the reports for details. According to the reviewers' reports, minor revisions still needed before your manuscript could be considered for acceptance. Please make sure that you respond to the reviewer's comments point-by-point. Please return your revised manuscript as soon as possible.

·

Basic reporting

Thank you for revising your manuscript and adding the suggested edits.

Experimental design

The revised manuscript shows clear and more transparent methods.

Validity of the findings

No comments. Revised articles meets standards.

Additional comments

Thank you for revising the manuscript with the suggested edits. Please consider adding the reasoning you provide in response to the comment as to why this pond was chosen as a sampling site in the main text of the article.

·

Basic reporting

I would to thank authors for the efforts to improve the manuscript, but it still requires multiple corrections.

Experimental design

pass

Validity of the findings

L405: Reference (Zhang, 2022) related to the different fish specie, please compare with studies of C. idella, if available (I’ve found some: https://doi.org/10.1016/j.aqrep.2023.101556 https://doi.org/10.1111/wej.12280 https://doi.org/10.1016/j.scitotenv.2020.142840). Moreover, Zhang et. al reports that “The sediment microbiota in the aquaculture system is one of the primary sources of microbiota in the fish, and the fish microbiota will also settle into the sediment microbiota; … many fish-derived microbes were observed in the sediment, among which 16 ± 1.6% were from gut microbiota, and 27 ± 3.1% were from gill microbiota.”
L447-L452: sentence is difficult to understand, please rephrase; what point is consistent with (Zheng et al., 2021) ?
L456: while the antibiotics resistance profiling is beyond the scope of present study, this part may be omitted. By the way “Some antibiotics, including enrofloxacin and florfenicol, are permitted and extensively used in the aquaculture industry” according to the literature: https://doi.org/10.3389/fmicb.2018.01284 https://doi.org/10.1186/s40168-019-0632-7 https://doi.org/10.1007/s11356-021-17825-4
As the Discussion section extended, there is still almost no comparison of the results with related studies. The Conclusion section is leaved unchanged. I believe that most of the readers read this section next to the Abstract. Please summarize the Results & Discussion section to highlight main point of the study.
My notes about the main text:
• Please carefully check the species names to be italicized, f.e. Candidatus Aquiluna (see https://lpsn.dsmz.de/genus/aquiluna-1) and so on
• L426: what [61] stands for?
• L439: word “correlate” is missing
• L456: “strict restrictions” please rephrase to avoid repetition

---

## Round 0.3 · accepted · Accept

After going through the point-by-point response letter, I believe that the authors have addressed all the reviewer's comments and concerns. The manuscript is now ready to be accepted.